# Evolving Populations of Diverse RL Agents with MAP-Elites

**Thomas Pierrot**
InstaDeep
t.pierrot@instadeep.com

**Arthur Flajolet**
InstaDeep
a.flajolet@instadeep.con

## Abstract

Quality Diversity (QD) has emerged as a powerful alternative optimization paradigm that aims at generating large and diverse collections of solutions, notably with its flagship algorithm MAP-ELITES (ME) which evolves solutions through mutations and crossovers. While very effective for some unstructured problems, early ME implementations relied exclusively on random search to evolve the population of solutions, rendering them notoriously sample-inefficient for high-dimensional problems, such as when evolving neural networks. Follow-up works considered exploiting gradient information to guide the search in order to address these shortcomings through techniques borrowed from either Black-Box Optimization (BBO) or Reinforcement Learning (RL). While mixing RL techniques with ME unlocked state-of-the-art performance for robotics control problems that require a good amount of exploration, it also plagued these ME variants with limitations common among RL algorithms that ME was free of, such as hyperparameter sensitivity, high stochasticity as well as training instability, including when the population size increases as some components are shared across the population in recent approaches. Furthermore, existing approaches mixing ME with RL tend to be tied to a specific RL algorithm, which effectively prevents their use on problems where the corresponding RL algorithm fails. To address these shortcomings, we introduce a flexible framework that allows the use of any RL algorithm and alleviates the aforementioned limitations by evolving populations of agents (whose definition include hyperparameters and all learnable parameters) instead of just policies. We demonstrate the benefits brought about by our framework through extensive numerical experiments on a number of robotics control problems, some of which with deceptive rewards, taken from the QD-RL literature. We open source an efficient JAX-based implementation of our algorithm in the QDax library [1].

## 1 Introduction

Drawing inspiration from natural evolution's ability to produce living organisms that are both diverse and high-performing through competition in different niches, Quality Diversity (QD) methods evolve populations of diverse solutions to solve an optimization problem. In contrast to traditional Optimization Theory, where the goal is to find one solution maximizing a given scoring function, QD methods explicitly use a mapping from solutions to a vector space, referred to as a *behavior descriptor space*, to characterize solutions and maintain a data structure, referred to as a *repertoire*, filled with high-performing solutions that cover this space as much as possible, in a process commonly referred to as *illumination*. This new paradigm has led to breakthroughs over the past decade in many domains ranging from robotics control to engineering design and games generation (Gaier et al., 2018; Sarkar & Cooper, 2021; Gravina et al., 2019; Cully & Demiris, 2018). There are a number of advantages to QD methods over standard optimization ones. Actively seeking and maintaining diversity in a population of solutions has proved to be an effective exploration strategy, by reaching high-performing regions through a series of stepping stones, when the fitness function has no particular structure (Gaier et al., 2019). Additionally, having at disposal a diverse set of high-performing solutions can be greatly beneficial to a decision maker (Lehman et al., 2020), for instance because the scoring function may fail to model accurately the reality (Cully et al., 2015).

---

[1] https://github.com/adaptive-intelligent-robotics/QDax

MAP-ELITES (Mouret & Clune, 2015) has emerged as one of the most widely used algorithm in the QD community for its simplicity and efficacy. It divides the behavior descriptor space into a discrete mesh of cells and strives to populate them all with solutions with matching behavior descriptors that maximize the fitness function as much as possible. This algorithm has been used in many applications with great success, such as developing controllers for hexapod robots that can adapt to damage in real time (Cully et al., 2015). However, just like many evolutionary algorithms, it struggles on problems with high-dimensional search spaces, such as when evolving controllers parametrized by neural networks, as it uses random mutations and crossovers to evolve the population.

The breakthroughs of Deep Reinforcement Learning in sequential decision making problems prompted a new line of work in the QD field to make the algorithms capable of dealing with deep neural network parametrizations. These new methods borrow techniques from either Black-Box Optimization (BBO) or Reinforcement Learning (RL) in order to exploit gradient information to guide the search. Methods based on BBO techniques (Colas et al., 2020; Conti et al., 2018) follow the approaches from earlier works on scaling evolutionary algorithms to neuro-evolution, such as Salimans et al. (2017); Stanley & Miikkulainen (2002), and empirically evaluate gradients w.r.t. the parameters by stochastically perturbing them by small values a number of times. Methods borrowing tools from RL, such as Nilsson & Cully (2021); Pierrot et al. (2022), exploit the Markov-Decision-Process structure of the problem and adapt off-policy RL algorithms, such as TD3 (Fujimoto et al., 2018), to evolve the population. This often entails adding additional components to the evolutionary algorithm (e.g. a replay buffer, critic networks, hyperparameters of the RL agent, ...) and methods differ along the way these components are managed. RL-based MAP-ELITES approaches have outperformed other MAP-ELITES variants, and even state-of-the art RL methods, on a variety of robotics control problems that require a substantial amount of exploration due to deceptive or sparse rewards. However, the introduction of RL components in MAP-ELITES has come with a number of downsides: (i) high sensibility to hyperparameters (Khadka et al., 2019; Zhang et al., 2021), (ii) training instability, (iii) high variability in performance, and perhaps most importantly (iv) limited parallelizability of the methods due to the fact that many components are shared in these methods for improved sample-efficiency. Furthermore, existing RL-based MAP-ELITES approaches are inflexibly tied to a specific RL algorithm, which effectively prevents their use on problems where the latter fails.

These newly-introduced downsides are particularly problematic as they are some of the main advantages offered by evolutionary methods that are responsible for their widespread use. These methods are notoriously trivial to parallelize and there is almost a linear scaling between the convergence speed and the amount of computational power available, as shown in Lim et al. (2022) for MAP-ELITES. This is all the more relevant with the advent of modern libraries, such as JAX (Bradbury et al., 2018), that seamlessly enable not only to distribute the computations, including computations taking place in the physics engine with BRAX (Freeman et al., 2021), over multiple accelerators but also to fully leverage their parallelization capabilities through automated vectorization primitives, see Lim et al. (2022); Flajolet et al. (2022); Tang et al. (2022). Evolutionary methods are also notoriously robust to the exact choice of hyperparameters, see Khadka et al. (2019), which makes them suited to tackle new problems. This is in stark contrast with RL algorithms that tend to require problem-specific hyperparameter tuning to perform well (Khadka et al., 2019; Zhang et al., 2021).

In order to overcome the aforementioned limitations of RL-based MAP-ELITES approaches, we develop a new MAP-ELITES framework that **1.** can be generically and seamlessly compounded with any RL agent, **2.** is robust to the exact choice of hyperparameters by embedding a meta-learning loop within MAP-ELITES, **3.** is trivial to scale to large population sizes, which helps alleviating stochasticity and training stability issues, without entering offline RL regimes a priori by independently evolving populations of entire agents (including all of their components, such as replay buffers) instead of evolving policies only and sharing the other components across the population. Our method, dubbed PBT-MAP-ELITES, builds on MAP-ELITES and combines standard isoline operators with policy gradient updates to get the best of both worlds. We evaluate PBT-MAP-ELITES when used with the SAC (Haarnoja et al., 2018) and TD3 (Fujimoto et al., 2018) agents on a set of five standard robotics control problems taken from the QD literature and show that it either yields performance on par with or outperforms state-of-the-art MAP-ELITES approaches, in some cases by a strong margin, while not being provided with hyperparameters tuned beforehand for these problems. Finally, we open source an efficient JAX-based implementation of our algorithm that combines the efficient implementation of PBT from Flajolet et al. (2022) with that of MAP-ELITES from Lim et al. (2022). We refer to these two prior works for speed-up data points compared to alternative implementations.

## 2 BACKGROUND

**Problem Definition.** We consider the problem of generating a repertoire of neural policies that are all high-performing for a given task while maximizing the diversity of policies stored in the repertoire. More formally, we consider a finite-horizon Markov Decision Process (MDP) $(\mathcal{S}, \mathcal{A}, \mathcal{R}, \mathcal{T})$, where $\mathcal{A}$ is the action space, $\mathcal{S}$ is the state space, $\mathcal{R} : \mathcal{S} \times \mathcal{A} \to \mathbb{R}$ is the reward signal, $\mathcal{T} : \mathcal{S} \times \mathcal{A} \to \mathcal{S}$ is the transition function, and $T$ is the episode length. A neural policy corresponds to a neural network $\pi_\theta : \mathcal{S} \to \mathcal{D}(\mathcal{A})$ where $\theta \in \Theta$ denotes the weights of the neural network and $\mathcal{D}(\mathcal{A})$ is the space of distributions over the action space. At each time step, we feed the current environment state to the neural network and we sample an action from the returned distribution, which we subsequently take. Once the action is carried out in the environment, we receive a reward and the environment transitions to a new state. The fitness $F(\pi_\theta)$ of a policy $\pi_\theta$ is defined as the expected value of the sum of rewards thus collected during an episode. We denote the space of trajectories thus followed in the environment by $\tau \in \Omega$. In the QD literature, diversity is not directly measured in the parameter space $\Theta$, but rather in another space $\mathcal{D}$, referred to as the behavior descriptor space or sometimes simply descriptor space, which is defined indirectly through a pre-specified and problem-dependent mapping $\Phi : \Omega \to \mathcal{D}$. A policy $\pi_\theta$ is thus characterized by rolling it out in the environment and feeding the trajectory to $\Phi$. With a slight abuse of notation, we denote by $\Phi(\pi_\theta)$ the behavior descriptor of the policy $\pi_\theta$. Diversity of a repertoire of policies is measured differently across QD approaches.

**MAP-Elites.** MAP-ELITES uses a tesselation technique to divide the descriptor space into a finite number of cells, which collectively define a discrete repertoire. In this work, we use the Centroidal Voronoi Tessellation (CVT) technique (Vassiliades et al., 2017) for all considered methods as it has been shown to be general and easy to use in practice (Vassiliades et al., 2017; Pierrot et al., 2022). MAP-ELITES starts by randomly initializing a set of $M$ policies. Each of these policies is then independently evaluated in the environment and they are sequentially inserted into the repertoire according to the following rule. If the cell corresponding to the descriptor of the policy at hand is empty, the policy is copied into this cell. In the opposite situation, the policy replaces the current incumbent only if it has a greater fitness and is dropped otherwise. During each subsequent iteration, policies are randomly sampled from the repertoire, copied, and perturbed to obtain a new set of $M$ policies which are then tentatively inserted into the repertoire following the aforementioned rule. Implementations of MAP-ELITES often differ along the exact recipe used to perturb the policies. The original MAP-ELITES algorithm (Mouret & Clune, 2015) relies on random perturbations. In this work, we use the isoline variation operator (Vassiliades & Mouret, 2018) that, given two parent policies, say policies $\theta_1$ and $\theta_2$, adds Gaussian noise $\mathcal{N}(0, \sigma_1)$ to $\theta_1$ and offsets the results along the line $\theta_2 - \theta_1$ by a magnitude randomly sampled from a zero-mean Gaussian distribution with variance $\mathcal{N}(0, \sigma_2)$. This strategy has proved to be particularly effective to evolve neural networks (Rakicevic et al., 2021). Pseudocode for MAP-ELITES is provided in the Appendix.

**BBO-based QD.** To improve sample efficiency and asymptotic performance, methods such as ME-ES (Colas et al., 2020) use first-order updates to perturb the policies with the objective of both increasing the fitness of the policies in the repertoire and improving the coverage of the repertoire (i.e. the number of non-empty cells). To generate the updates, ME-ES use the Evolution Strategy from Salimans et al. (2017). Specifically, after selecting a policy from the repertoire, its neural network parameters are perturbed stochastically with a small amount of Gaussian noise a number of times and the resulting policies are rolled out in the environment for a full episode. All of the collected samples are then used to empirically estimate gradients for a smoothed version around the starting policy of either (1) the fitness function, (2) a novelty function which is defined as the average Euclidean distance between the starting policy's behavior descriptor and its $k$ nearest neighbors among all previously computed behavior descriptors, or (3) alternatively the fitness function and the novelty function to increase both quality and diversity, which is the version we use in this work (see the Appendix for the pseudocode). Note that similar strategies using the NS-ES family of algorithms exist, such as Conti et al. (2018), but these methods are outperformed by ME-ES (Colas et al., 2020).

**RL-based QD.** Using evolution strategies to guide the search with first-order updates improves upon random search but remains doomed to a low sample-efficiency due to the need of rolling out a significant number of policies over entire trajectories to get reasonably accurate gradient estimates. More recent techniques, such as QD-PG (Pierrot et al., 2022) and PGA-MAP-ELITES (Nilsson & Cully, 2021), exploit the MDP structure of the problem and leverage policy-gradient techniques from RL as well as off-policy extensions for improved sample efficiency and better asymptotic convergence.

Both QD-PG and PGA-MAP-ELITES build on the TD3 agent (Fujimoto et al., 2018). PGA-MAP-ELITES combines random mutations derived through the isoline variation operator with mutations obtained through policy gradient computations. QD-PG introduces the notion of a diversity reward, a signal defined at the timestep-level to drive policies towards unexplored regions of the behavior descriptor space, which makes it possible to leverage the RL machinery to compute policy gradients to increase the diversity of the population, referred to as diversity policy gradients, in addition to the standard policy gradients to increase the fitness of the policies, referred to as quality policy gradients. At each MAP-ELITES iteration, half of the selected policies are updated using quality policy gradients and the other half are updated using diversity policy gradients. In contrast to PGA-MAP-ELITES, QD-PG does not relies on random search updates. Both QD-PG and PGA-MAP-ELITES use a single shared replay buffer where all the transitions collected when evaluating the agents are stored and from which batches are sampled to compute policy gradients.

Critic networks are managed differently by each algorithm. QD-PG uses two different sets of critic parameters, one for quality rewards and one for diversity rewards, that are shared across the population and both are updated any time a policy gradient is computed. PGA-MAP-ELITES maintains a greedy policy and its associated critic which are updated independently of the rest of the repertoire. The greedy policy is regularly inserted in the repertoire and the critic is used to compute policy gradients updates for all other policies but is only updated using the greedy policy.

These precise design choices not only make PGA-MAP-ELITES and QD-PG difficult to distribute efficiently but they also harm the flexibility of these methods. For instance, if one would like to replace TD3 by another popular off-policy algorithm such as SAC, which is known to perform better for some environments, numerous new design choices arise. For instance for SAC, one would have to decide how to handle the temperature parameter and the entropy target within the population. Furthermore, while sharing critic parameters and using a single replay buffer was motivated by a desire for greater sample efficiency, this introduces new issues when scaling these methods. For instance, as the number of policies updated concurrently at each iteration increases we get closer to an offline RL setting, which is known to harm performance, since all policies share the same replay buffer. Conversely, as the size of the repertoire increases, any single policy stored in the repertoire is updated all the less frequently than the critic which may cause them to significantly lag behind over time. Finally, both QD-PG and PGA-MAP-ELITES assume that good hyperparameters are provided for TD3 while it is known that tuning these values for the problem at hand is necessary to get good performance. This effectively puts the burden on the user to tune hyperparameters for TD3 as a preliminary step, which limits the usability of such methods in new settings. Pseudocodes for QD-PG and PGA-MAP-ELITES are provided in the Appendix.

## 3 METHOD

In order to overcome the limitations of RL-based QD methods identified in the last section, we revisit the neuro-evolution problem defined in Section 2 and introduce a new algorithm, dubbed PBT-MAP-ELITES, that evolves populations of agents as opposed to populations of policies. An agent is defined by a tuple $(\theta, \phi, \mathbf{h})$ where $\theta$ denotes the policy parameters, $\phi$ denotes all other learnable parameters of the agent (e.g. critic parameters and target critic parameters), and $\mathbf{h}$ denotes its hyperparameters (e.g. learning rates and magnitude of the exploration noise). As in the original formulation, we assume that the fitness and behavior descriptor functions depend only on the policy, i.e. on $\theta$. The learnable parameters and the hyperparameters are only used when agents are updated. PBT-MAP-ELITES internally uses a policy-search-based RL algorithm which can be selected freely by the user. In particular, it may be on-policy or off-policy.

PBT-MAP-ELITES maintains a MAP-ELITES repertoire as well as a population of $P$ agents. The population is randomly initialized (including the hyperparameters), evaluated, copied and inserted into the repertoire. We also initialize $P$ replay buffers if the underlying RL algorithm makes use of them. Additionally, a batch of agents is sampled from the repertoire and a variation operator is applied to obtain $M$ offspring that are also evaluated and inserted into the repertoire as part of the initialization phase. Then, the algorithm proceeds in iterations, each of which consists of two consecutive steps: 1. population update and 2. MAP-ELITES repertoire update.

**Population Update.** To update the population of agents, we use the following strategy inspired from Jaderberg et al. (2017). We first rank all agents in the population by fitness based on the evaluation

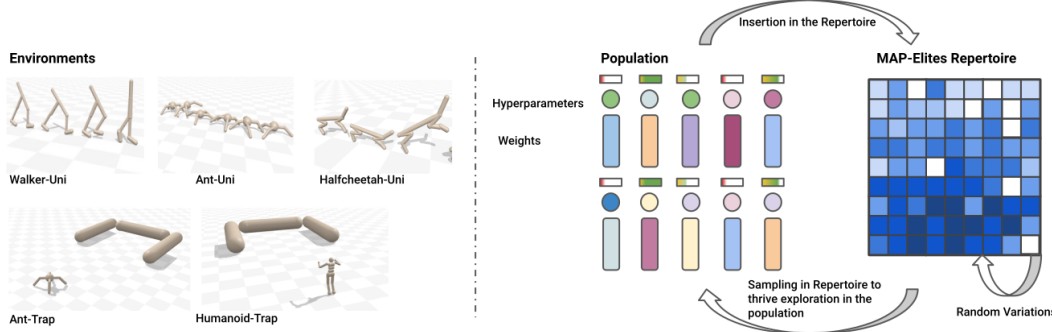

Figure 1: **Left Panel.** The set of QD-RL environments used to evaluate all methods. All environments are implemented using the BRAX simulator and are available in the QDAX suite. In WALKER2D-UNI, ANT-UNI, and HALFCHEETAH-UNI, the goal is to learn to control the corresponding robot to run as fast as possible with gaits that are as diverse as possible. In ANT-TRAP and HUMANOID-TRAP, the robot should travel as far as possible along the x-axis direction, which entails sidestepping the trap to get the highest possible return thus putting the exploration capabilities of considered methods to test. **Right panel.** Illustration of the PBT-MAP-ELITES algorithm. PBT-MAP-ELITES maintains a population of agents (policy parameters, other learnable parameters as well as hyperparameters) and a MAP-ELITES repertoire of agents. The repertoire update, generating more diversity, and the population update, improving the fitness of the policies, are intertwined.

that took place at the end of the last iteration. Agents that are in the bottom $p\%$ of the population are replaced by agents sampled uniformly from the top $n\%$ of the population, with $0 < p < 1 - n < 1$. We also randomly select $k\%$ of the agents in the population among the ones that are neither in the top $n\%$ nor in the bottom $p\%$ and we replace them by agents randomly sampled from the current MAP-ELITES repertoire. All other agents remain unchanged. This mechanism allows potentially lower-performing, but more diverse, individuals from the repertoire to enter the population while maintaining high-performing agents alive. When agents are replaced, new hyperparameter values are sampled uniformly from pre-specified ranges. The agents' policy parameters as well as all other learnable parameters are subsequently trained for $S$ steps, using the user-selected RL algorithm. If needed, the collected experience is stored inside the replay buffers. In contrast to PGA-MAP-ELITES and QD-PG, we closely follow the general recipe followed by most RL algorithms and only add the experience collected during training, in exploration mode, to the replay buffers while the experience collected during evaluation, in exploitation mode, is discarded. Additionnaly, note that the agents are trained independently from one another, which makes it trivial to parallelize the most computationally intensive part of this step. This is in stark contrast with other MAP-ELITES-RL methods that share some parameters across the population, e.g. the critic parameters for QD-PG and PGA-MAP-ELITES, which are typically updated concurrently by all agents.

**Repertoire Update.** Once the agents in the population have been trained, they are evaluated and inserted into the repertoire. Then, just like during the initialization phase, a batch of agents is randomly sampled from the repertoire and undergoes a variation operator to obtain $M$ offspring which are evaluated and inserted into the grid. As in PGA-MAP-ELITES, the variation operator is meant to increase the descriptor space coverage but we have also observed that this process stabilizes the algorithm as a whole. In order to define a variation operator that can be used with agents, as opposed to policies, we deal with variations over the policy and learnable parameters separately from variations over the hyperparameters. Specifically, an isoline operator is applied to policy and other learnable parameters while the offspring simply inherit the hyperparameters of one of their parents. While more sophisticated strategies could be investigated, we have observed that this simple mechanism works well in practice in our experiments.

Observe that optimization of the quality as well as the diversity of the policies happens at two different levels in PBT-MAP-ELITES. Quality is encouraged through both the elitist population update and the repertoire insertion mechanism. Diversity is induced through both the addition of agents from the repertoire to the population and the use of random variation operators at each iteration. The pseudocode of the algorithm is provided in the Appendix.

## 4  LITERATURE REVIEW

**Quality Diversity.** QD methods aim to simultaneously maximize diversity and performance. Among existing options, MAP-ELITES and Novelty Search with Local Competition (NSLC) are two of the most popular QD algorithms. NSLC builds on the Novelty Search algorithm (Lehman & Stanley, 2011) and maintains an unstructured archive of solutions selected for their high performance relative to other solutions in their neighborhoods while MAP-ELITES relies on a tesselation technique to discretize the descriptor space into cells. Both algorithms rely extensively on Genetic Algorithms (GA) to evolve solutions. As a result, they struggle when the dimension of the search space increases, which limits their applicability. These approaches have since been extended using tools from Evolution Strategies (ES) to improve sample efficiency and asymptotic performance over the original implementations based on GA (Salimans et al., 2017). CMA-MAP-ELITES (Fontaine et al., 2020) relies on Covariance Matrix Adaptation (CMA) to speed up the illumination of the descriptor space. NSRA-ES and NSR-ES (Conti et al., 2018) build on recent ES tools to improve QD methods' exploration capabilities on deep RL problems with deceptive or sparse rewards. ME-ES (Colas et al., 2020) introduces alternate ES updates for quality and diversity in order to solve deep RL problems with continuous action spaces that require a good amount of exploration. While ES-based approaches improve over GA-based ones, they are still relatively sample-inefficient due to the fact that they need to roll out a large of number of policies over entire trajectories to empirically estimate gradients with reasonable accuracy. Several recent methods propose to exploit analytical gradients when this is possible instead of estimating them empirically. DQD (Fontaine & Nikolaidis, 2021) builds a mutation operator that first computes gradients of the fitness and behavior descriptor functions at the current solution and carry out a first-order step by summing the gradients with random coefficients. Tjanaka et al. (2022) applies the same technique to deep RL problems with continuous action spaces. PGA-MAP-ELITES (Nilsson & Cully, 2021) and QD-PG (Pierrot et al., 2022) exploit the MDP structure of the problems to compute policy gradients using the TD3 algorithm, outperforming all QD competitors for deep RL problems with continuous actions. However, both methods are tied a single RL algorithm and are highly sensitive to the choice of TD3 hyperparameters.

**Population Based Reinforcement Learning.** Our work has different motivations than classical RL algorithms as we do not aim to find a policy than achieves the best possible return but rather to illuminate a target descriptor space. However, we share common techniques with Population-Based RL (PBRL) algorithms. In this field, the closest method to ours is the Population-Based-Training (PBT) algorithm (Jaderberg et al., 2017) which uses a genetic algorithm to learn the hyperparameters of a population of RL agents concurrently to training them. While PBT-MAP-ELITES and PBT use similar strategies to update the population of agents, PBT only seeks the highest-performing agent by extracting the best one from the final population while PBT-MAP-ELITES aims to find a diverse collection of high-performing agents. Several methods such as CERL, ERL, and CEM-RL (Pourchot & Sigaud, 2019; Khadka & Tumer, 2018; Khadka et al., 2019) combine ES algorithms with PBRL methods to improve the asymptotic performance and sample efficiency of standard RL methods. Other methods, such as DVD (Parker-Holder et al., 2020) and P3S-TD3 (Jung et al., 2020), train populations of agents and add terms in their loss functions to encourage the agents to explore different regions of the state-action space but always with the end goal of maximizing the performance of the best agent in the population. Flajolet et al. (2022) show how to vectorize computations across the population to run PBRL algorithms as efficiently as possible on accelerators through the use of the JAX library. Lim et al. (2022) introduced similar techniques to accelerate MAP-ELITES through the evaluation of thousands of solutions in parallel with JAX. In this study, we build on both of these works and implement PBT-MAP-ELITES in the JAX framework to make it fast and scalable.

## 5  EXPERIMENTS

**Environments.** We use five robotics environments that fall into two categories:

**1.** HALFCHEETAH-UNI, WALKER2D-UNI and ANT-UNI are environments widely used in the QD community to evaluate an algorithm's ability to illuminate a complex descriptor space, see for instance Cully et al. (2015); Nilsson & Cully (2021); Tjanaka et al. (2022). In these environments, the goal is to make a legged robot run as fast as possible along the forward direction while optimizing for diversity w.r.t. the robot's gaits, indirectly characterized as the mean frequencies of contacts between the robots' legs and the ground. This last quantity defines the behavior descriptor for these environ-

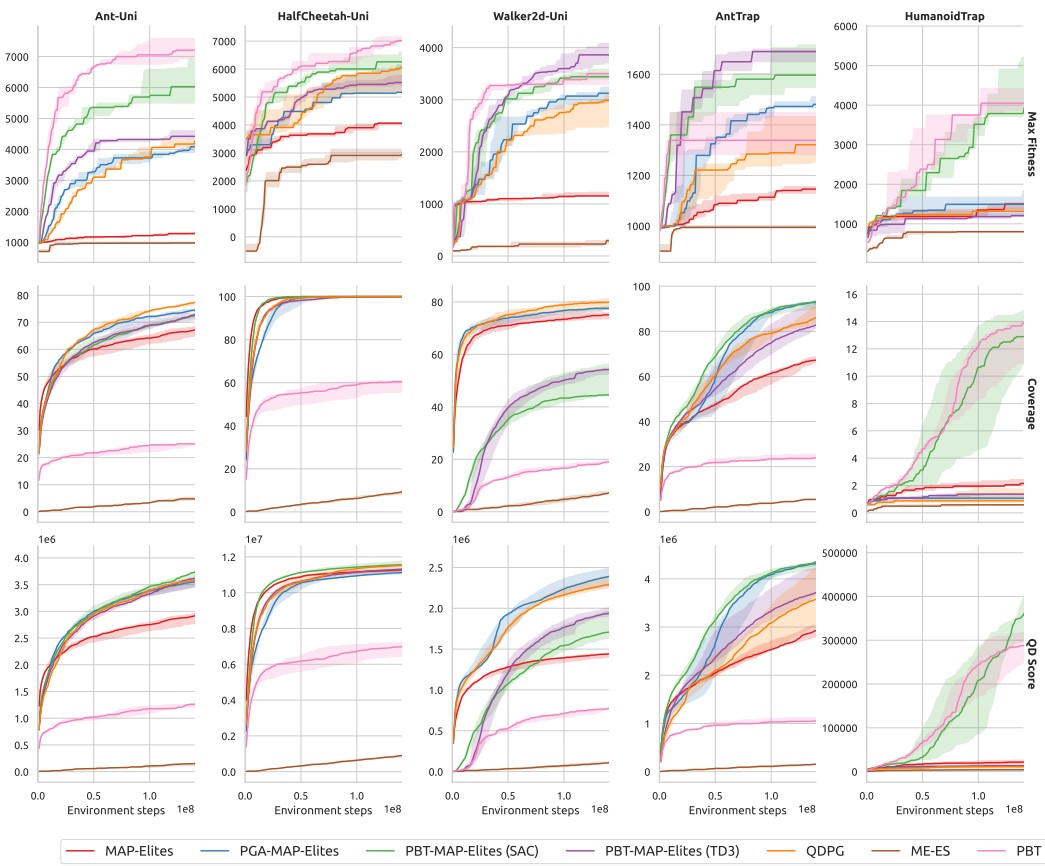

Figure 2: Performance comparison of PBT-MAP-ELITES with baselines on the basis of standard metrics from the QD literature for five environments from the QDAX suite (which is based on the BRAX engine). We benchmark two variants of PBT-MAP-ELITES, one where it is composed with SAC and one where it is composed with TD3. All methods are trained with a total budget of $N = 1.5e8$ environment timesteps. Experiments are repeated over 5 runs with different random seeds and the medians (resp. first and third quartile intervals) are depicted with full lines (resp. shaded areas).

ments while the reward at each timestep is the velocity of the robot's center of gravity projected onto the forward direction.

**2.** ANT-TRAP and HUMANOID-TRAP are environments with deceptive reward signals used in the QD-RL literature to evaluate an algorithm's ability to solve complex continuous control problems that require a good amount of exploration, see Colas et al. (2020); Conti et al. (2018); Pierrot et al. (2022). In these environments, the goal is also to make the legged robot run as fast as possible in the forward direction, though with the additional difficulty that the robot is initially facing a trap. As a result, following the reward signal in a greedy fashion leads the robot into the trap. The robot must explore the environment and learn to go around the trap, even though this is temporarily suboptimal, in order to obtain higher returns. In these environments, the behavior descriptor is defined as the position of the robot's center of gravity at the end of an episode. All of these environments are based on the BRAX simulator (Freeman et al., 2021) and are available in the QDAX suite (Lim et al., 2022).

**Setup.** We compare PBT-MAP-ELITES to state-of-the-art MAP-ELITES-based methods, namely MAP-ELITES, ME-ES, PGA-MAP-ELITES as well as QD-PG. For these experiments, we benchmark two variants of PBT-MAP-ELITES: one where it is composed with SAC and one where it is composed with TD3. For the sake of fairness, we use the same values for parameters that are used by multiple methods. In particular, all MAP-ELITES-based methods maintain a repertoire of 1024 cells and use CVT with the same parametrization to discretize the behavior descriptor space into 1024 cells. Similarly, when a variation operator is needed, we always use the isoline operator with the same pa-

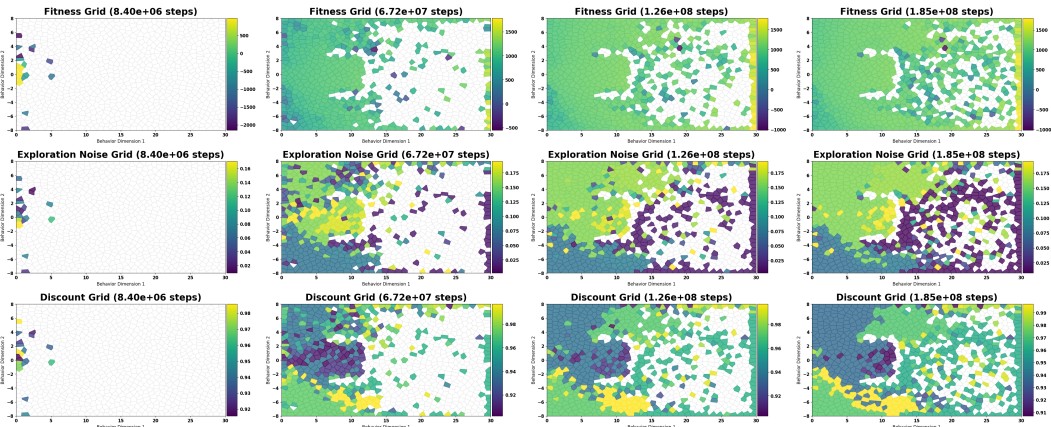

Figure 3: Visualizations of the performance and hyperparameters of the agents stored in the MAP-ELITES repertoire of PBT-MAP-ELITES (TD3) at different time points during training on the ANT-TRAP environment. The first row corresponds to the fitnesses of the agents. The second row corresponds to the exploration noise added by the agents to the actions returned by their policies when collecting experience in exploration mode. The third row corresponds to the discount factor $\gamma$ used by the agents. Snapshots of the repertoire are taken at uniformly-spaced intervals during training.

rameters $\sigma_1 = 0.005$ and $\sigma_2 = 0.05$. All policy and critic networks are implemented by two MLPs layers with 256 hidden neurons per layer. For methods relying on the TD3 agent, the hyperparameters used are the ones introduced in the original paper for MUJOCO environments. Pseudocodes and parameter values for all algorithms under study are provided in the Appendix.

Additionally, we compare PBT-MAP-ELITES to the PBT algorithm (Jaderberg et al., 2017) (pseudocode provided in the Appendix) when it is used to optimize populations of SAC agents. Both PBT-MAP-ELITES and PBT evolve populations of 80 agents and use the same ranges for the hyperparameters. All policy and critic networks are implemented by two-layer MLPs with 256 hidden neurons per layer, just like for TD3 for PGA-MAP-ELITES and QD-PG. Furthermore, the parameters of all agents in the population are identically initialized. For PBT-MAP-ELITES (resp. PBT), agents in the bottom $p = 0.2$ (resp. $p = 0.4$) fraction of the population (in terms of fitness) are replaced by agents sampled from the top $n = 0.1$ fraction of the population. For PBT-MAP-ELITES, a fraction $k = 0.4$ of the agents that are neither in the bottom $20\%$ nor in the top $10\%$ of the population are replaced by agents randomly sampled from the MAP-ELITES repertoire. All other parameters and design choices are identical for these two methods.

**Metrics and fair comparisons.** Following standard practice in the QD literature, we monitor three metrics used to evaluate the performance of a collection of policies during training. 1. We measure the **maximum fitness**, defined as the maximum expected return across policies in the collection. 2. We measure the **coverage** over the descriptor space, computed as the number of cells that have been filled. 3. We measure the **QD-score**, computed as the sum of fitnesses attained by the policies stored in the repertoire. For this last metric to be meaningful, we assume that fitnesses are all non-negative. If not, a positive value is added to all fitnesses to enforce it. In any case, this value is the same for all methods for fairness. Since some of the metrics require a repertoire to be properly defined, we introduce a passive repertoire for PBT to be able to evaluate it on the same basis as the other methods. Specifically, at the end of each PBT iteration, the population of agents generated by PBT is evaluated and inserted into a repertoire. For each method, we report the evolution of these metrics w.r.t. the total number of interactions with the environment. Note that while the evaluation of an agent contributes to the total number of interactions for MAP-ELITES-based methods, this is not the case for PBT as the evaluations are only used to estimate the metrics for this method.

## 6 RESULTS AND DISCUSSION

Statistics on QD metrics are reported for all environments and methods on Figure 2.

**Performance comparison to other MAP-ELITES-based methods.** We observe that PBT-MAP-ELITES (SAC) is the only method able to solve HUMANOID-TRAP within the allocated timestep budget, outperforming all the other methods by a significant margin. HUMANOID-TRAP is a challenging environment as obtaining high returns requires not only to get the humanoid robot to run, which is a challenging continuous problem in itself, but also to learn to sidestep the trap in spite of a deceptive reward signal. This environment, introduced in Colas et al. (2018), has remained out of reach for MAP-ELITES-based methods, setting aside ME-ES which solves it with a timestep budget two orders of magnitude higher. Interestingly, the maximum fitness remains below 2000 for TD3-based methods, which means they were not able to get the humanoid robot to run at all. This is a testament to the difficulty of the problem. Recall that TD3 was not able to solve the MUJOCO-based version of the Humanoid environment in the original paper that introduced this algorithm (Fujimoto et al., 2018). A careful tuning of the algorithm design choices and hyperparameters, carried out in a later study, was required to get TD3 to perform well on this environment. Setting aside the WALKER2D-UNI environment, note that PBT-MAP-ELITES (SAC) either outperforms, often by a significant margin for the maximum fitness metric, or performs on par with MAP-ELITES-based methods. Interestingly, the SAC variant of PBT-MAP-ELITES often performs better than the TD3 variant, but not always. On a side note, we also observe that ME-ES surprisingly gets outperformed by all MAP-ELITES competitors, including the original MAP-ELITES algorithm, in all environments. This can be explained by the fact that ME-ES uses 1000 evaluations (i.e. $1e6$ timesteps) to update a single policy. As a result, for a repertoire consisted of 1024 cells and with a budget of $1.5e8$ timesteps, the maximum coverage that can be reached by ME-ES is 15% only. In the original study, ME-ES manages to outperform other MAP-ELITES-based methods with a budget of $1e10$ timesteps.

**Performance comparison to PBT.** We observe that PBT outperforms the SAC variant of PBT-MAP-ELITES in terms of maximum fitness on HALFCHEETAH-UNI and ANT-UNI. This is expected as: (1) these environments do not require a significant amount of exploration, (2) PBT only aims to maximize the maximum fitness, and (3) PBT-MAP-ELITES aims to maximize both the maximum fitness and the policies' diversity. However, we observe the opposite trend on ANT-TRAP and HUMANOID-TRAP where significant exploration is required to achieve high returns given the deceptive nature of the reward signal. We conclude that optimizing for diversity turns out to play a crucial role for these two environments. As expected, PBT-MAP-ELITES outperforms PBT in terms of coverage and QD-score in all environments, setting aside HUMANOID-TRAP. The seemingly unexpected results observed on HUMANOID-TRAP stem from the fact that covering the behavior descriptor directly correlates with exploration of the $(x, y)$ space, which is required to achieve high returns in this environment due to the presence of the trap.

**Repertoire interpretation.** By visualizing the evolution of the fitnesses and hyperparameters of the agents stored in PBT-MAP-ELITES's repertoire at different time points during training, see Figure 3, we observe that PBT-MAP-ELITES evolves locally-coherent (w.r.t. the descriptor space) maps of hyperparameters that change significantly during training. In particular, we remark that PBT-MAP-ELITES dynamically increases the amount of exploration noise of the TD3 agents to boost exploration when needed to go around the trap and decreases this parameter once the trap has been sidestepped to focus on getting high returns. This mechanism gives a significant advantage to PBT-MAP-ELITES over QD-PG and PGA-MAP-ELITES, for which this parameter is set to a constant value.

## 7  CONCLUSION

In this work, we revisit the standard formulation of the QD neuro-evolution problem by evolving repertoires of full agents (including hyperparameters among other things) as opposed to only policies. This extension brings flexibility compared to existing frameworks as it allows us to combine any RL algorithm with MAP-ELITES in a generic and scalable fashion. This formulation also allows us to dynamically learn the hyperparameters of the underlying RL agent as part of the regular training process, which removes a significant burden from the user. Surprisingly, we observe that learning the hyperparameters improves both the asymptotic performance and the sample efficiency in practice for most of the environments considered in this work. Our method is the first to solve the HUMANOID-TRAP environment with less than one billion interactions with the simulator, to be compared with tens of billions of interactions for state-of-the-art QD methods. We hope that this work constitutes one more step towards bridging the gap between Neuro-Evolution and Reinforcement Learning, combining the best of both worlds in a simple framework.

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

# A    PSEUDOCODES FOR ALL ALGORITHMS

---

**Algorithm 1:** PBT-MAP-ELITES

---

**Given:**

- $\mathbb{M}$: MAP-ELITES repertoire
- $N \in \mathbb{N}^*$: maximum number of environment steps
- $M \in \mathbb{N}^*$: number of isoline-variation offsprings per iteration
- $P \in \mathbb{N}^*$: size of the population of RL agents
- $S \in \mathbb{N}^*$: number of training steps per iteration per agent
- $p, k, n \in ]0, 1[$: PBT proportions
- an RL agent template
- $F(\cdot)$: fitness function
- $\Phi(\cdot)$: behavior descriptor function

```
// Initialization
```
Randomly initialize $P + M$ agents following the chosen RL template $((\pi_{\theta_i}, \phi_i, \mathbf{h}_i))_{1 \leq i \leq P+M}$.
Run one episode in the environment using each of $(\pi_{\theta_i})_{1 \leq i \leq P+M}$ to evaluate $(F(\pi_{\theta_i}))_{1 \leq i \leq P+M}$ and
  $(\Phi(\pi_{\theta_i}))_{1 \leq i \leq P+M}$.
Insert $((\pi_{\theta_i}, \phi_i, \mathbf{h}_i))_{1 \leq i \leq P+M}$ in $\mathbb{M}$ based on $(F(\pi_{\theta_i}))_{1 \leq i \leq P+M}$ and $(\Phi(\pi_{\theta_i}))_{1 \leq i \leq P+M}$.
Initialize $P$ replay buffers $(\mathbb{B}_i)_{1 \leq i \leq P}$ using the data collected respectively by each agent during the initial
  evaluations (if replay buffers are used by the RL agent).

```
// Main loop
```
Initialize $n_{\text{steps}}$, the total number of environment interactions carried out so far, to 0.
**while** $n_{steps} \leq N$ **do**

$\quad$
```
// Population Update
```
$\quad$ Re-order the agents $i = 1, \cdots, P$ in increasing order of their fitnesses $(F(\pi_{\theta_i}))_{1 \leq i \leq P}$.
$\quad$ Update agents $i = 1, \cdots, pP$ by copying randomly-sampled agents from $i = (1 - n)P, \cdots, P$ and
$\quad\quad$ copy the replay buffers accordingly (if replay buffers are used by the RL agent).
$\quad$ Sample new hyperparameters for agents $i = 1, \cdots, pP$.
$\quad$ Sample $kP$ indices $(i_j)_{1 \leq j \leq kP}$ uniformly without replacement from $\{pP + 1, \cdots, (1 - n)P - 1\}$.
$\quad$ Replace agents $i = i_j, 1 \leq j \leq kP$ by agents randomly(-uniformly) sampled from $\mathbb{M}$.
$\quad$ Train agents $i = 1, \cdots, P$ independently for $S$ steps using the RL agent template, sampling data
$\quad\quad$ from the replay buffers if they are used by the RL agent.

$\quad$
```
// Repertoire Update
```
$\quad$ Run one episode in the environment using each of $(\pi_{\theta_i})_{1 \leq i \leq P}$ to evaluate $(F(\pi_{\theta_i}))_{1 \leq i \leq P}$ and
$\quad\quad$ $(\Phi(\pi_{\theta_i}))_{1 \leq i \leq P}$.
$\quad$ Insert $((\pi_{\theta_i}, \phi_i, \mathbf{h}_i))_{1 \leq i \leq P}$ in $\mathbb{M}$ based on $(F(\pi_{\theta_i}))_{1 \leq i \leq P}$ and $(\Phi(\pi_{\theta_i}))_{1 \leq i \leq P}$.
$\quad$ Sample uniformly $2M$ agents from $\mathbb{M}$.
$\quad$ Copy them and apply isoline variation to obtain $M$ offsprings $((\pi_{\theta_i}, \phi_i, \mathbf{h}_i))_{P < i \leq P+M}$.
$\quad$ Run one episode in the environment using each of $(\pi_{\theta_i})_{P < i \leq P+M}$ to evaluate $(F(\pi_{\theta_i}))_{P < i \leq P+M}$
$\quad\quad$ and $(\Phi(\pi_{\theta_i}))_{P < i \leq P+M}$.
$\quad$ Insert $((\pi_{\theta_i}, \phi_i, \mathbf{h}_i))_{P < i \leq P+M}$ in $\mathbb{M}$ based on $(F(\pi_{\theta_i}))_{P < i \leq P+M}$ and $(\Phi(\pi_{\theta_i}))_{P < i \leq P+M}$.
$\quad$ Update $n_{\text{steps}}$.

**end**

---

---

**Algorithm 2:** MAP-ELITES

---

**Given:**

- $\mathbb{M}$: MAP-ELITES repertoire
- $N \in \mathbb{N}^*$: maximum number of environment steps
- $M \in \mathbb{N}^*$: number of offsprings per iteration
- $F(\cdot)$: fitness function
- $\Phi(\cdot)$: behavior descriptor function

```
// Initialization
```
Randomly initialize $M$ policies $(\pi_{\theta_i})_{1 \leq i \leq M}$.
Run one episode in the environment using each of $(\pi_{\theta_i})_{1 \leq i \leq M}$ to evaluate $(F(\pi_{\theta_i}))_{1 \leq i \leq M}$ and $(\Phi(\pi_{\theta_i}))_{1 \leq i \leq M}$.
Insert $(\pi_{\theta_i})_{1 \leq i \leq M}$ in $\mathbb{M}$ based on $(F(\pi_{\theta_i}))_{1 \leq i \leq M}$ and $(\Phi(\pi_{\theta_i}))_{1 \leq i \leq M}$.

```
// Main loop
```
Initialize $n_{\text{steps}}$, the total number of environment interactions carried out so far, to 0.
**while** $n_{steps} \leq N$ **do**

    Randomly sample $2M$ policies from $\mathbb{M}$.
    Copy them and apply isoline variations to obtain $M$ new policies $(\pi_{\theta_i})_{1 \leq i \leq M}$.
    Run one episode in the environment using each of $(\pi_{\theta_i})_{1 \leq i \leq M}$ to evaluate $(F(\pi_{\theta_i}))_{1 \leq i \leq M}$ and $(\Phi(\pi_{\theta_i}))_{1 \leq i \leq M}$.
    Insert $(\pi_{\theta_i})_{1 \leq i \leq M}$ in $\mathbb{M}$ based on $(F(\pi_{\theta_i}))_{1 \leq i \leq M}$ and $(\Phi(\pi_{\theta_i}))_{1 \leq i \leq M}$.
    Update $n_{\text{steps}}$.

**end**

---

---

**Algorithm 3:** PGA-MAP-ELITES

---

**Given:**

- $\mathbb{M}$: MAP-ELITES repertoire
- $N \in \mathbb{N}^*$: maximum number of environment steps
- $M \in \mathbb{N}^*$: number of offsprings per iteration
- $S_c \in \mathbb{N}^*$: number of TD3 training steps used to update the shared critic per iteration
- $S_p \in \mathbb{N}^*$: number of TD3 policy update steps per iteration per policy
- TD3 hyperparameters
- $F(\cdot)$: fitness function
- $\Phi(\cdot)$: behavior descriptor function

```
// Initialization
```
Initialize a replay buffer $\mathbb{B}$.
Randomly initialize $M$ policies $(\pi_{\theta_i})_{1 \leq i \leq M}$.
Run one episode in the environment using each of $(\pi_{\theta_i})_{1 \leq i \leq M}$ to evaluate $(F(\pi_{\theta_i}))_{1 \leq i \leq M}$ and
  $(\Phi(\pi_{\theta_i}))_{1 \leq i \leq M}$.
Insert $(\pi_{\theta_i})_{1 \leq i \leq M}$ in $\mathbb{M}$ based on $(F(\pi_{\theta_i}))_{1 \leq i \leq M}$ and $(\Phi(\pi_{\theta_i}))_{1 \leq i \leq M}$.
Update $\mathbb{B}$ with transition data collected during the initial evaluations.
Initialize the critic $Q_\phi$, the target critic $Q_{\phi'}$, the greedy policy $\pi_\theta$, and the target greedy policy $\pi_{\theta'}$.

```
// Main loop
```
Initialize $n_{\text{steps}}$, the total number of environment interactions carried out so far, to 0.
**while** $n_{\text{steps}} \leq N$ **do**

  ```
  // Update the shared critic alongside the greedy policy
  ```
  Carry out $S_c$ TD3 training steps to update $Q_\phi, Q_{\phi'}, \pi_\theta$ and $\pi_{\theta'}$ (sampling batches of data from $\mathbb{B}$).

  ```
  // Generate new offsprings using the isoline variation operator
  ```
  Randomly sample $M$ policies from $\mathbb{M}$.
  Copy them and apply isoline variations to obtain $M/2$ new policies $(\pi_{\theta_i})_{1 \leq i \leq M/2}$.

  ```
  // Generate new offsprings using TD3 policy-gradient updates
  ```
  Randomly sample $M/2 - 1$ policies from $\mathbb{M}$ $(\pi_{\theta_i})_{M/2 < i \leq M-1}$.
  Carry out $S_p$ TD3 policy gradient steps for each of them independently (sampling batches of data
    from $\mathbb{B}$).

  ```
  // Update the repertoire
  ```
  Assign $\pi_{\theta_M} = \pi_\theta$.
  Run one episode in the environment using each of $(\pi_{\theta_i})_{1 \leq i \leq M}$ to evaluate $(F(\pi_{\theta_i}))_{1 \leq i \leq M}$ and
    $(\Phi(\pi_{\theta_i}))_{1 \leq i \leq M}$.
  Insert $(\pi_{\theta_i})_{1 \leq i \leq M}$ in $\mathbb{M}$ based on $(F(\pi_{\theta_i}))_{1 \leq i \leq M}$ and $(\Phi(\pi_{\theta_i}))_{1 \leq i \leq M}$.
  Update $\mathbb{B}$ with transition data collected during the evaluations of all $M$ new policies.
  Update $n_{\text{steps}}$.

**end**

---

---

**Algorithm 4:** ME-ES

---

**Given:**

- $\mathbb{M}$: MAP-ELITES repertoire
- $N \in \mathbb{N}^*$: maximum number of environment steps
- $S \in \mathbb{N}^*$: number of consecutive gradient steps for a given policy
- $N_{\mathbf{grad}} \in \mathbb{N}^*$: number of evaluations for gradient approximations
- $N_{\mathbf{init}} \in \mathbb{N}^*$: number of randomly-initialized policies used to initialize $\mathbb{M}$
- $\sigma > 0$: standard deviation of the normal distribution used to perturb parameters for gradient approximations
- $\eta > 0$: learning rate
- $\mathbb{A}$: archive of behavior descriptors
- $N(\cdot, \cdot)$: novelty function that takes as an input a behavior descriptor as first argument and $\mathbb{A}$ as a second argument
- $F(\cdot)$: fitness function
- $\Phi(\cdot)$: behavior descriptor function

```
// Initialization
```
Randomly initialize $N_{\mathbf{init}}$ policies $(\pi_{\theta_i})_{1 \le i \le N_{\mathbf{init}}}$.
Run one episode in the environment using each of $(\pi_{\theta_i})_{1 \le i \le N_{\mathbf{init}}}$ to evaluate $(F(\pi_{\theta_i}))_{1 \le i \le N_{\mathbf{init}}}$ and
  $(\Phi(\pi_{\theta_i}))_{1 \le i \le N_{\mathbf{init}}}$.
Insert $(\pi_{\theta_i})_{1 \le i \le N_{\mathbf{init}}}$ in $\mathbb{M}$ based on $(F(\pi_{\theta_i}))_{1 \le i \le N_{\mathbf{init}}}$ and $(\Phi(\pi_{\theta_i}))_{1 \le i \le N_{\mathbf{init}}}$.
Add $(\Phi(\pi_{\theta_i}))_{1 \le i \le N_{\mathbf{init}}}$ to $\mathbb{A}$.

```
// Main loop
```
Initialize $n_{\text{steps}}$, the total number of environment interactions carried out so far, to 0.
Initialize $n_{\text{grads}}$, the total number of gradient steps carried out so far, to 0.
use_novelty = true
**while** $n_{steps} \le N$ **do**

    **if** $n_{grads} \equiv 0 \mod S$ **then**
        ```// Decide if we should optimize for novelty or fitness.```
        Set use_novelty to true with probability 0.5 and to false otherwise.

        ```// Sample a high-performing policy from M```
        **if** *use_novelty* **then**
            Sample a policy $\pi_\theta \in \mathbb{M}$ uniformly from the set of five policies with the highest novelty
                $N(B(\pi_\theta), \mathbb{A})$.
        **else**
            Sample, with probability 0.5, a policy $\pi_\theta \in \mathbb{M}$ from the set of two policies with the highest
                fitness $F(\pi_\theta)$ or from the last five updated policies.
        **end**
    **end**

    ```// Update the current policy using a gradient approximation```
    Sample $(\theta_i)_{1 \le i \le N_{\mathbf{grad}}} \sim \mathcal{N}(\theta, \sigma^2 I)$ small perturbations of the current policy's parameters.
    Run one episode in the environment using each of the corresponding policies $(\pi_{\theta_i})_{1 \le i \le N_{\mathbf{grad}}}$ to
      evaluate $(F(\pi_{\theta_i}))_{1 \le i \le N_{\mathbf{grad}}}$ and $(\Phi(\pi_{\theta_i}))_{1 \le i \le N_{\mathbf{grad}}}$.
    **if** *use_novelty* **then**
        Compute the gradient approximation $\nabla\theta = \frac{1}{N_{\mathbf{grad}}\sigma} \sum_{i=1}^{N_{\mathbf{grad}}} N(\Phi(\pi_{\theta_i}), \mathbb{A}) \frac{\theta_i - \theta}{\sigma}$.
    **else**
        Compute the gradient approximation $\nabla\theta = \frac{1}{N_{\mathbf{grad}}\sigma} \sum_{i=1}^{N_{\mathbf{grad}}} F(\pi_{\theta_i}) \frac{\theta_i - \theta}{\sigma}$.
    **end**
    Update $\theta = \theta + \eta \cdot \nabla\theta$.
    Run one episode in the environment using $\pi_\theta$ to compute $\Phi(\pi_\theta)$ and $F(\pi_\theta)$.
    Insert $\pi_\theta$ in $\mathbb{M}$ based on $\Phi(\pi_\theta)$ and $F(\pi_\theta)$.
    Add $\Phi(\pi_\theta)$ to $\mathbb{A}$.
    Update $n_{\text{steps}}$.
    $n_{\text{grads}} = n_{\text{grads}} + 1$
**end**

---

---

**Algorithm 5:** QD-PG

---

**Given:**

- $\mathbb{M}$: MAP-ELITES repertoire
- $N \in \mathbb{N}^*$: maximum number of environment steps
- $P \in \mathbb{N}^*$: size of the population of RL agents
- $S \in \mathbb{N}^*$: number of TD3 training steps per iteration per agent
- TD3 hyperparameters
- $N(\cdot)$: novelty reward function
- $F(\cdot)$: fitness function
- $\Phi(\cdot)$: behavior descriptor function

```
// Initialization
```
Initialize a replay buffer $\mathbb{B}$.
Randomly initialize P policies $(\pi_{\theta_i})_{1 \leq i \leq P}$.
Run one episode in the environment using each of $(\pi_{\theta_i})_{1 \leq i \leq P}$ to evaluate $(F(\pi_{\theta_i}))_{1 \leq i \leq P}$ and
   $(\Phi(\pi_{\theta_i}))_{1 \leq i \leq P}$.
Insert $(\pi_{\theta_i})_{1 \leq i \leq P}$ in $\mathbb{M}$ based on $(F(\pi_{\theta_i}))_{1 \leq i \leq P}$ and $(\Phi(\pi_{\theta_i}))_{1 \leq i \leq P}$.
Update $\mathbb{B}$ with transition data collected during the initial evaluations.
Initialize the *quality* (resp. *diversity*) critic $Q_\phi^Q$ (resp. $Q_\phi^D$) and the corresponding target $Q_{\phi'}^Q$ (resp. $Q_{\phi'}^D$).

```
// Main loop
```
Initialize $n_{\text{steps}}$, the total number of environment interactions carried out so far, to 0.
**while** $n_{steps} \leq N$ **do**

    Sample uniformly $P$ policies $(\pi_{\theta_i})_{1 \leq i \leq P}$ from $\mathbb{M}$.

    ```
// Update the quality critic alongside the first half of the
   policies
```
    **for** $s = 1$ **to** $S$ **do**
        Sample $P/2$ batches of transitions from $\mathbb{B}$.
        Carry out, using one batch of transition per agent, one TD3 training step for each of the agents
           $((\pi_{\theta_i}, Q_\phi^Q, Q_{\phi'}^Q))_{1 \leq i \leq P/2}$ in parallel, averaging gradients over the agents for the shared critic
           parameters.
    **end**

    ```
// Update the diversity critic alongside the second half of the
   policies
```
    **for** $s = 1$ **to** $S$ **do**
        Sample $P/2$ batches of transitions from $\mathbb{B}$.
        Overwrite the rewards using the novelty reward function $N(\cdot)$.
        Carry out, using one batch of transition per agent, one TD3 training step for each of the agents
           $((\pi_{\theta_i}, Q_\phi^D, Q_{\phi'}^D))_{P/2 < i \leq P}$ in parallel, averaging gradients over the agents for the shared critic
           parameters.
    **end**

    ```
// Update the repertoire
```
    Run one episode in the environment using each of $(\pi_{\theta_i})_{1 \leq i \leq P}$ to evaluate $(F(\pi_{\theta_i}))_{1 \leq i \leq P}$ and
       $(\Phi(\pi_{\theta_i}))_{1 \leq i \leq P}$.
    Insert $(\pi_{\theta_i})_{1 \leq i \leq P}$ in $\mathbb{M}$ based on $(F(\pi_{\theta_i}))_{1 \leq i \leq P}$ and $(\Phi(\pi_{\theta_i}))_{1 \leq i \leq P}$.
    Update $\mathbb{B}$ with transition data collected during the evaluations of all $P$ new policies.
    Update $n_{\text{steps}}$.
**end**

---

---

**Algorithm 6:** PBT

---

**Given:**

- $N \in \mathbb{N}^*$: maximum number of environment steps
- $P \in \mathbb{N}^*$: size of the population of RL agents
- $S \in \mathbb{N}^*$: number of training steps per iteration per agent
- $p, n \in \,]0, 1[$: PBT proportions
- an RL agent template
- $F(\cdot)$: fitness function

```
// Initialization
```
Randomly initialize $P$ agents following the chosen RL template $((\pi_{\theta_i}, \phi_i, \mathbf{h}_i))_{1 \le i \le P}$.
Initialize $P$ replay buffers $(\mathbb{B}_i)_{1 \le i \le P}$ (only if replay buffers are used by the RL agent).

```
// Main loop
```
Initialize $n_{\text{steps}}$, the total number of environment interactions carried out so far, to 0.
**while** $n_{steps} \le N$ **do**

   Train agents $i = 1, \cdots, P$ independently for $S$ steps using the RL agent template and the replay buffers (only if replay buffers are used by the RL agent), interacting with the environment as many times as dictated by the RL agent.
   Run one episode in the environment using each of $(\pi_{\theta_i})_{1 \le i \le P}$ to evaluate $(F(\pi_{\theta_i}))_{1 \le i \le P}$.
   Re-order the agents $i = 1, \cdots, P$ in increasing order of their fitnesses $(F(\pi_{\theta_i}))_{1 \le i \le P}$.
   Update agents $i = 1, \cdots, pP$ by copying randomly-sampled agents from $i = (1 - n)P, \cdots, P$ and copy the replay buffers accordingly (only if replay buffers are used by the RL agent).
   Sample new hyperparameters for agents $i = 1, \cdots, pP$.
   Update $n_{\text{steps}}$.

**end**

---

# B  EXPERIMENTAL DETAILS

In this section, we detail the parameters used for all algorithms. In particular, we stress that we use the same values used in the original studies for all MAP-ELITES-based algorithms other than the one introduced in this paper, namely MAP-ELITES, PGA-MAP-ELITES, QD-PG, and ME-ES. Additionally, we run the implementations of these algorithms provided in the QDAX library Lim et al. (2022) for our experiments. All MAP-ELITES-based algorithms use a grid with 1024 cells initialized using CVT with 50,000 initial random points.

Table 1: PBT parameters

| Parameter | Value |
|---|---|
| Population size $P$ | 80 |
| Proportion of worst agents $p$ | 0.4 |
| Proportion of best agents $n$ | 0.1 |
| Number of training steps per iteration per agent $S$ | 5000 |
| Replay buffer size | 100000 |

Table 2: PBT-MAP-ELITES parameters

| Parameter | Value |
|---|---|
| Number of isoline-variation offsprings per iteration $M$ | 240 |
| Size of the population of RL agents $P$ | 80 |
| Proportion of worst agents $p$ | 0.2 |
| Proportion of agents to sample from the repertoire $k$ | 0.4 |
| Proportion of best agents $n$ | 0.1 |
| Number of training steps per iteration per agent $S$ | 5000 |
| Replay buffers size | 100000 |
| Isoline $\sigma_1$ | 0.005 |
| Isoline $\sigma_2$ | 0.05 |

Table 3: MAP-ELITES parameters.

| Parameter | Value |
|---|---|
| Number of offsprings per iteration $M$ | 1000 |
| Isoline $\sigma_1$ | 0.005 |
| Isoline $\sigma_2$ | 0.05 |

Table 4: ME-ES parameters.

| Parameter | Value |
|---|---|
| Number of consecutive gradient steps for a given policy $S$ | 10 |
| Number of evaluations for gradient approximations $N_{\mathbf{grad}}$ | 1000 |
| Number of randomly-initialized policies used to initialize the repertoire $N_{\mathbf{init}}$ | 1 |
| Std of the normal distribution to perturb parameters for gradient approximations $\sigma$ | 0.2 |
| Learning rate $\eta$ | 0.01 |

Table 5: PGA-MAP-ELITES parameters.

| Parameter | Value |
|---|---|
| Number of offsprings per iteration $M$ | 100 |
| Number of TD3 training steps used to update the shared critic per iteration $S_c$ | 300 |
| Number of TD3 policy update steps per iteration per policy $S_p$ | 100 |
| Discount factor $\gamma$ | 0.99 |
| Policy learning rate | 3e-4 |
| Critic learning rate | 3e-4 |
| Noise clipping | 0.5 |
| Policy noise | 0.2 |
| Exploration noise | 0.0 |
| Soft update tau | 0.005 |
| Batch size | 256 |
| Replay buffer size | 100000 |

Table 6: QD-PG parameters.

| Parameter | Value |
|---|---|
| Size of the population of RL agents $P$ | 100 |
| Number of TD3 training steps per iteration per agent $S$ | 100 |
| Discount factor $\gamma$ | 0.99 |
| Policy learning rate | 3e-4 |
| Critic learning rate | 3e-4 |
| Noise clipping | 0.5 |
| Policy noise | 0.2 |
| Exploration noise | 0.0 |
| Soft update tau | 0.005 |
| Batch size | 256 |
| Replay buffer size | 100000 |

Table 7: SAC hyperparameters' ranges (or values if the hyperparameter does not change during training) that PBT and PBT-MAP-ELITES sample from.

| Parameter | Range / Value |
|---|---|
| Discount factor $\gamma$ | [0.9, 1.0] |
| Policy learning rate | [3e-5, 3e-3] |
| Critic learning rate | [3e-5, 3e-3] |
| Alpha learning rate | [3e-5, 3e-3] |
| Reward scaling factor | [0.1, 10] |
| Soft update tau | 0.005 |
| Alpha initial value | 1.0 |
| Batch size | 256 |

Table 8: TD3 hyperparameters' ranges (or values if the hyperparameter does not change during training) that PBT and PBT-MAP-ELITES sample from.

| Parameter | Range / Value |
|---|---|
| Discount factor $\gamma$ | [0.9, 1.0] |
| Policy learning rate | [3e-5, 3e-3] |
| Critic learning rate | [3e-5, 3e-3] |
| Noise clipping | [0.0, 1.0] |
| Policy noise | [0.0, 1.0] |
| Exploration noise | [0.0, 0.2] |
| Soft update tau | 0.005 |
| Batch size | 256 |

