# OpenReview forum: "Evolving Populations of Diverse RL Agents with MAP-Elites"
_ICLR.cc/2023/Conference — ICLR 2023 poster_

### Official Review · Reviewer_Y8Hf · 2022-10-23

**Confidence:** 3
**Correctness:** 3
**Technical Novelty And Significance:** 3
**Empirical Novelty And Significance:** 2
**Recommendation:** 6

**Clarity, Quality, Novelty And Reproducibility:**

- Clarity: Overall clear. The text is readable, but awkwardly structured: long discussions of literature occupy the first 3 pages of the text before introducing the core problem and contributions.
- Quality: Good. The results show reasonable performance and success on hard exploration tasks. The experiments demonstrate rigor for fair comparison.
- Originality: Modest. The novelty in the work lies in the design decisions made for representing policies (by including hyperparameters, replay buffers, &c. in agents) and allowing two processes to modify policies.
- Reproducibility: Modest (possibly high?). The appendix provides algorithm details and the text is clear about the major experiment and algorithm design features. No code is provided for the implementation, but the text describes open sourcing the implementation so this may be forthcoming.

**Strength And Weaknesses:**

## Strengths

1. Comparable performance to existing algorithms while being able to solve hard exploration tasks (HumanoidTrap, AntTrap) fast (in terms of environment steps).
2. Generality. The approach presented can readily be coupled with many RL algorithms and directly addresses hyperparameter sensitivity issues in those algorithms. It would be interesting to see results showing how this helps compared to baselines on other RL tasks.


## Weaknesses

1. Few major performance improvements over alternatives. The results would be stronger if there were clear environments that previous algorithms simply failed to solve that PBT-MAP-Elites solves. The closest result is the HumanoidTrap result. This is not a major weakness.
2. No data on scaling. One of the main benefits of the approach is that it "3) can scale to large population sizes", but this is never tested. Adding experiments showing this scaling and how it compares to alternatives would greatly benefit the paper.
3. No data on (wallclock) speedup. Can the speedup of running experiments in JAX be quantified? The JAX implementation is claimed as being efficient, but no empirical evidence backs this claim compared to other simulators.


## Feedback & Questions
- Figure 2: The colors for ME-ES and PBT are hard to distinguish, particularly when shaded areas overlap the medians. Consider slightly different colors for readability. Also it would help to order the columns to have all "*Uni" columns and "*Trap" columns grouped together.
- Figure 3: The axis text is too small to read. Please make it larger.
- Note: At least two references are duplicated: Pierrot 2022a & 2022b, Lim 2022a & 2022b.
- What is the speedup of using the JAX implementation compared to alternatives?
- The introduction and related work are both quite lengthy and cover overlapping material. Consider condensing the introduction to highlight the main contributions and novelty, with the related work discussing the particular limitations of past efforts. This can help keep the narrative clear for readers and focus attention on the specific novelty of the work reported.

**Summary Of The Paper:**

The paper proposes a quality-diversity algorithm for reinforcement learning that enables both the RL optimization step and population repertoire update step to make changes to agent parameters. The proposed algorithm allows updates to agent hyperparameters to reduce the sensitivity of the approach to hyperparameter choice of the underlying RL algorithm. Scaling is achieved by allowing parallel evolution of agent populations and a JAX implementation for high performance. Results show strong performance (matching competitive alternatives) on robotic control tasks, particularly in terms of deceptive environments and when evaluated not only on final performance but also coverage of strategies.

**Summary Of The Review:**

The technical novelty of the paper is modifying MAP-Elites to use a PBT-like structure for the agent population. This provides the benefit of generality to RL implementation and automatic tuning of hyperparameters (inherited from PBT). This is certainly new, though incremental in the sense that both PBT and MAP-Elites have been thoroughly explored in the past. There is substantial benefit to be had from further investigation of this kind of combination of RL meta-algorithms (like PBT and self-play) with evolutionary approaches, particularly from quality diversity.

Empirical results show some strong results in the set of tasks the algorithms are tailored to: deceptive domains. Most tasks show competitive performance with alternatives. All results are in relatively simple robotic control tasks (compared to controlling robotic manipulators or real robot locomotion), leaving plenty of room for stronger tests of the algorithm capabilities. No empirical results examine the speedup claims of the JAX implementation or the scalability claims for the algorithm, though both seem very plausible.

Taken together the work makes a step forward for quality-diversity approaches to RL in deceptive continuous control tasks. Whether this approach has wider value remains an open question.

---

> ### Author Response · Authors · 2022-11-19
> **Response to Reviewer Y8Hf**
>
> - Few major performance improvements over alternatives. The results would be stronger if there were clear environments that previous algorithms simply failed to solve that PBT-MAP-Elites solves. The closest result is the HumanoidTrap result. This is not a major weakness.
>
> We believe that Humanoid-Trap serves this purpose as this is a very challenging problem that, as demonstrated on Figure 2, all previous MAP-Elites-based algorithms considered in this work fail at (in the sense that they fail to get past the trap). To the best of our knowledge, only the authors of QD-PG report some success on this problem but only for some seeds whereas the performance reported on Figure 2 is consistent with the fact that PBT-MAP-Elites (SAC) gets past the trap every time.
>
> - No data on scaling. One of the main benefits of the approach is that it "3) can scale to large population sizes", but this is never tested. Adding experiments showing this scaling and how it compares to alternatives would greatly benefit the paper.
>
> We meant to say that the method is (i) easier to scale from a practical standpoint as the RL agents are trained independently (whereas some parameters are co-trained by all RL agents in QD-PG and PGA-MAP-Elites) and (ii) should avoid offline reinforcement learning regimes for the same reasons. We now mention this explicitly in the main body of the paper. That said, we agree that additional experiments would be needed to truly demonstrate the scaling power of the approach compared to alternatives. However this would likely require introducing new environments given that the ones considered in this work are already solved by PBT-MAP-Elites.
>
> - No data on (wallclock) speedup. Can the speedup of running experiments in JAX be quantified? The JAX implementation is claimed as being efficient, but no empirical evidence backs this claim compared to other simulators.
>
> We rely on the JAX implementations released as part of the QDax library, see Lim et al. (2022), for the MAP-Elites component and the environments and the JAX implementations of PBT-TD3 and PBT-SAC released alongside Flajolet et al. (2022). There are extensive numerical experiments in these two prior works quantifying the speedups that can be obtained depending on the hardware used. We now refer to these two works in the main body of this paper for concrete speedup data points.
>
> - Figure 2: The colors for ME-ES and PBT are hard to distinguish, particularly when shaded areas overlap the medians. Consider slightly different colors for readability. Also it would help to order the columns to have all "*Uni" columns and "*Trap" columns grouped together.
>
> Thank you for your suggestions. We have grouped all "*Uni" columns together and all "*Trap" columns together. We have also used a different panel of colors that should make it much easier to distinguish one method from another.
>
> - Note: At least two references are duplicated: Pierrot 2022a & 2022b, Lim 2022a & 2022b.
>
> Thank you for pointing this out. We have removed duplicated references
>
> - What is the speedup of using the JAX implementation compared to alternatives?
>
> This question has been addressed as part of another response above.
>
> - The introduction and related work are both quite lengthy and cover overlapping material. Consider condensing the introduction to highlight the main contributions and novelty, with the related work discussing the particular limitations of past efforts. This can help keep the narrative clear for readers and focus attention on the specific novelty of the work reported.
>
> Thank you for your suggestion. We have shortened the abstract, introduction, and background sections, removing overlapping sections, to leave more room to describe the algorithm introduced in this work as well as the numerical experiments.
>
> - Reproducibility: Modest (possibly high?). The appendix provides algorithm details and the text is clear about the major experiment and algorithm design features. No code is provided for the implementation, but the text describes open sourcing the implementation so this may be forthcoming.
>
> We have taken three steps to make it much easier to reproduce our results. First, we have now included a clean version of the code used to run experiments in the supplementary material. This code will be publicly released after the publication of this work. Second, we provide a fully anonymized Colab notebook to run our algorithm on any of the robotics control problems considered in this work. Armed with this notebook, anybody can run the code without having to install any dependency. Third, we have included a detailed account of the parameters used for all algorithms under study in Section B of the Appendix.

---

### Official Review · Reviewer_XjfY · 2022-10-25

**Confidence:** 4
**Correctness:** 2
**Technical Novelty And Significance:** 2
**Empirical Novelty And Significance:** 3
**Recommendation:** 5

**Clarity, Quality, Novelty And Reproducibility:**

The paper is easy to follow in general. However, because the algorithm is written only by the natural language, it is hard to understand precisely. I see the novelty in the proposed framework including hyper-parameter learning mechanism. However, the approach is relatively a straight-forward extension of existing MAP-ELITES approaches. The experimental details are provided in the appendix.

**Strength And Weaknesses:**

# Strength

Approach: The approach is relatively simple. The RL baseline can be easily replaced. And it is parallel implementation friendly.

Evaluation: The advantages of the proposed approach over existing MAP-ELITES approaches have been demonstrated on robotics environments in terms of the best fitness, coverage, and QD score. In particular, the advantage is pronounced on environment with deceptive reward signals is reported.

# Weaknesses

Clarity: The algorithm is described thoroughly by natural language in the main text. It is not easy to precisely understand, in particular, for those who are not familiar with MAP-ELITES. The algorithm is provided in the appendix, but the details are not given there. Moreover, because of a different symbol used in the algorithm and in the main text (M vs N), it was hard to fully understand.

Algorithm validity: The authors say that the meta-learning mechanism for the hyper-parameter is included in the algorithm. However, as far as I understand, the hyper-parameter is only randomly sampled uniformly over the domain. whenever an agent in the population is replaced. It does not look like optimizing the hyper-parameter values.

Motivation and Algorithm Design and Experimental Evaluation: The authors highlighted four difficulties (written in the summary part) and proposed the approach to address these difficulties. However, it was not clear from the paper how they are addressed. Item (i) seems to be addressed by the meta-learning of the hyper-parameter. However, as I wrote above, it was not clear why this makes sense. I couldn't find discussion related to Item (ii) and (iii). The experiments are not designed to evaluate these points.

**Summary Of The Paper:**

This paper addresses the limitations of reinforcement learning (RL) based MAP-ELITES approaches, an approach to quality diversity (QD) optimization. The authors highlight the following four problems with existing approaches: (i) high sensibility to hyperparameters; (ii) training instability; (iii) high variability in performance; and (iv) limited parallelizability.

To overcome these limitations, the authors propose a population-based RL-based MAP-EILTES approach. Each agent in the population encodes the policy parameters, the hyper-parameters for the training, and some other internal parameters used during the training. The proposed framework is a generic framework in that one can easily replace the RL baseline to train each agent.

The proposed approach is compared with other MAP-Elites variants with SAC and TD3 as their RL baselines on five robotics environments that are often used in QD-RL literature. Superior performance, in particular on environments with deceptive reward signals, are observed.

**Summary Of The Review:**

I see the advantages of the proposed approach. However, because of the lack of the discussion of the algorithm validity and (empirical) analysis of the algorithmic behavior, I think that the claims are not sufficiently supported in the paper.

---

> ### Author Response · Authors · 2022-11-19
> **Response to Reviewer XjfY**
>
> - Clarity: The algorithm is described thoroughly by natural language in the main text. It is not easy to precisely understand, in particular, for those who are not familiar with MAP-ELITES. The algorithm is provided in the appendix, but the details are not given there. Moreover, because of a different symbol used in the algorithm and in the main text (M vs N), it was hard to fully understand.
>
> We have now added pseudocodes for all of the algorithms under study in Section A of the Appendix to make the paper more self-contained. The consistent use of notations across pseudocodes to refer to the same components should also help identify patterns and spot differences between existing algorithms and the algorithm introduced in this paper. We have also fixed all notational inconsistencies throughout the main body and the appendix.
>
> - Algorithm validity: The authors say that the meta-learning mechanism for the hyper-parameter is included in the algorithm. However, as far as I understand, the hyper-parameter is only randomly sampled uniformly over the domain. whenever an agent in the population is replaced. It does not look like optimizing the hyper-parameter values.
>
> The fact that hyperparameters are optimized over effectively during the course of the algorithm can be seen on Figure 3. There are two mechanisms through which hyperparameters are optimized in our algorithm.
>
> First, hyperparameters are re-sampled only for the worst performing elements of the population of RL agents at fixed training intervals while the best performing agents have their hyperparameters unchanged. As a result, at the end of each iteration, the best hyperparameters found so far are always kept while we also inject new hyperparameter values through randomly sampling for the worst performing agents. This mechanism is largely inspired from the hyperparameter-optimization algorithm introduced in Jaderberg et al. (2017), PBT, which is currently widely used.
>
> Second, agents are inserted and removed from the MAP-Elites repertoire depending on their performances. Agents that are doomed to perform poorly in the environment because of a poor choice of hyperparameters will eventually be removed from the repertoire.
>
> - Motivation and Algorithm Design and Experimental Evaluation: The authors highlighted four difficulties (written in the summary part) and proposed the approach to address these difficulties. However, it was not clear from the paper how they are addressed. Item (i) seems to be addressed by the meta-learning of the hyper-parameter. However, as I wrote above, it was not clear why this makes sense. I couldn't find discussion related to Item (ii) and (iii). The experiments are not designed to evaluate these points.
>
> The meta-learning of the hyperparameters portion of the question is addressed above.
>
> While we agree that we did not design experiments to specifically evaluate training instability (ii) and high variability in performance (iii), both of these conditions affect an algorithm’s ability to perform well on average and this translates into poor average performance on any - reasonably hard - problem. We believe that average performance is what matters the most to practitioners. Hence, (ii) and (iii) are captured indirectly in our experiments, though we agree that it is likely that they are compounded with other factors.
>
> - However, the approach is relatively a straight-forward extension of existing MAP-ELITES approaches.
>
> Paraphrasing our answer to Reviewer bFfK: We believe that this is a major advantage of the method we have developed. For instance, the popularity of MAP-Elites and PBT is a result of the effectiveness and parallelizability of these methods just as much as of their simplicity. Most, if not all, of the existing methods combining RL with MAP-Elites are quite complex with intricate details that make them hard to reproduce (and parallelize). We believe that PBT-MAP-ELITES has the potential to feel a gap in this field.

---

### Official Review · Reviewer_bFfK · 2022-11-02

**Confidence:** 3
**Correctness:** 3
**Technical Novelty And Significance:** 2
**Empirical Novelty And Significance:** 2
**Recommendation:** 5

**Clarity, Quality, Novelty And Reproducibility:**

The paper is clear but not always precise and sufficiently detailed. I do not think is possible to replicate the results. Too many details are missing.

**Strength And Weaknesses:**

Strengths:
The changes made to MAP-ELITES are simple but seem to be effective.
Weaknesses:
- the PBT-MAP-ELITES algorithm is a variation of PGA-MAP-ELITES, which seems quite simple even though it appears to be effective.
- There are many variations of MAP-ELITES that are used in the experiments but for which there is no explanation in the paper.
- The experimental results included are relatively limited and are not described with enough details to be reproducible.
- The paper assumes familiarity with all the specific robotics problems used to test the algorithm.
- The figure with the experimental results (Fig.2) is hard to read.

**Summary Of The Paper:**

The paper presents a framework that allows using any reinforcement learning (RL) algorithm within a population of agents. The contribution is to use quality diversity methods to evolve populations and maintain their diversity. The most commonly used method is MAP-ELITES, but MAP-ELITES does not work well on high-dimensional search spaces. The contribution of the paper is the development of a version of MAP-ELITES, called PBT-MAP-ELITES, that does not depend on a specific RL agent, is robust to hyperparameter choices, and scales to large population sizes.  Some experimental results are included for an example problem.

**Summary Of The Review:**

The paper presents an algorithm that is a new variation of MAP-ELITES. It evolves a population of agents, scales well to large population sizes, and is robust to hyperparameter choices. The experimental results that are included in the paper show good performance compared to other algorithms, but they are described in a succinct way, making it very hard to replicate the results.

---

> ### Author Response · Authors · 2022-11-19
> **Response to Reviewer bFfK**
>
> - the PBT-MAP-ELITES algorithm is a variation of PGA-MAP-ELITES, which seems quite simple even though it appears to be effective.
>
> We believe that this is a major advantage of the method we have developed. For instance, the popularity of MAP-Elites and PBT is a result of the effectiveness and parallelizability of these methods just as much as of their simplicity. Most, if not all, of the existing methods combining RL with MAP-Elites are quite complex with intricate details that make them hard to reproduce (and parallelize). We believe that PBT-MAP-ELITES has the potential to feel a gap in this field.
>
>
> - There are many variations of MAP-ELITES that are used in the experiments but for which there is no explanation in the paper.
>
> We agree that we did not provide enough details about MAP-Elites variants from the literature taken as baselines in this work. We have now added pseudocodes for all of them in Section A of the Appendix to make the paper more self-contained. The consistent use of notations across pseudocodes to refer to the same components should also help identify patterns and spot differences.
>
> - The experimental results included are relatively limited and are not described with enough details to be reproducible.
>
> We have taken three steps to make it much easier to reproduce our results. First, we have now included a clean version of the code used to run experiments in the supplementary material. Second, we provide a fully-anonymized Colab notebook to run our algorithm on any of the robotics control problems considered in this work. Armed with this notebook, anybody can run the code without having to install any dependency. Third, we have included a detailed account of the parameters used for all algorithms under study in Section B of the Appendix.
>
> http address of the Colab notebook: https://colab.research.google.com/drive/10uoDdN1O03aThqnHuAuCbSPdgELVVxVP?usp=sharing
>
> - The paper assumes familiarity with all the specific robotics problems used to test the algorithm.
>
> We agree that, setting aside short descriptions at the beginning of Section 5, we do not delve into the specifics of the robotic control problems. These robotics problems are widely used in the QD community, see Cully et al. (2015); Nilsson & Cully (2021); Tjanaka et al. (2022); Colas et al. (2020); Conti et al. (2018); Pierrot et al. (2022), and are based on robotics control environments used extensively in the RL literature so we decided to defer to these prior works given the space constraint. That said, we did provide an illustration of the environments we use on the left panel of Figure 1 so that readers unfamiliar with these problems can get an idea of what they are.
>
> - The figure with the experimental results (Fig.2) is hard to read.
>
> We have taken three steps to improve the readability of Figure 2. First, we use a different panel of colors that are easier to distinguish from one another. Second, we have increased the size of the figure. Third, we have switched to using a pdf format, which makes it possible to zoom in at will.

---

### Author Response · Authors · 2022-11-19
**General response to reviewers**

We would like to thank the reviewers for their suggestions that will surely improve the paper. We answer each reviewer independently point by point but first we would like to point out that, in order to improve the reproducibility of this work, we now provide: (i) a fully-anonymized Colab notebook to run our algorithm on any of the robotics control problems considered in this work (http address below) and (ii) a clean version of the code used to run all experiments in the supplementary material.

https://colab.research.google.com/drive/10uoDdN1O03aThqnHuAuCbSPdgELVVxVP?usp=sharing

---

### Decision · Program_Chairs · 2023-01-20

**Decision:**

Accept: poster

**Justification For Why Not Higher Score:**

The algorithm is not novel enough for a spotlight.

**Justification For Why Not Lower Score:**

Good paper, well written, simple algorithm with some novelty, good results.

**Metareview: Summary, Strengths And Weaknesses:**

The paper presents a new variation on Map-Elites, PBT-MAP-ELITES, and shows that it works well for finding sets of policies in reasonably high-dimensional spaces. The new algorithm is quite simple; the actual contributions are in the selection and update, and the algorithm is oblivious as to which RL algorithm to use as underlying algorithm. While the novelty is not overwhelming, I think this is a nice and useful enough framework. The results are also good, narrowly outperforming all competitors at several standard benchmarks (although I would have liked to see a comparison to Tjanaka's DQD-RL).

**Note From Pc:**

if the above contains the word "oral" or "spotlight" please see: "oral" presentation means -> notable-top-5% and "spotlight" means -> notable-top-25%. As stated in our emails, we are disassociating presentation type from AC recommendations